

# Composition, taxonomy and functional diversity of the oropharynx microbiome in individuals with schizophrenia and controls

Eduardo Castro-Nallar[1,6], Matthew L. Bendall[1],
Marcos Pérez-Losada[1,5,7], Sarven Sabuncyan[2], Emily G. Severance[2],
Faith B. Dickerson[3], Jennifer R. Schroeder[4], Robert H. Yolken[2] and
Keith A. Crandall[1]

[1] Computational Biology Institute, George Washington University, Ashburn, VA, USA
[2] Stanley Neurovirology Laboratory, Johns Hopkins School of Medicine, Baltimore, MD, USA
[3] Sheppard Pratt Hospital, Baltimore, MD, USA
[4] Schroeder Statistical Consulting LLC, Ellicott City, MD, USA
[5] CIBIO-InBIO, Centro de Investigação em Biodiversidade e Recursos Genéticos, Universidade do Porto, Vairão, USA
[6] Center for Bioinformatics and Integrative Biology, Universidad Andrés Bello, Facultad de Ciencias Biológicas, Santiago, Chile
[7] Division of Emergency Medicine, Children's National Medical Center, Washington, D.C., USA

Corresponding author
Eduardo Castro-Nallar,
castronallar@gmail.com

## ABSTRACT

The role of the human microbiome in schizophrenia remains largely unexplored. The microbiome has been shown to alter brain development and modulate behavior and cognition in animals through gut-brain connections, and research in humans suggests that it may be a modulating factor in many disorders. This study reports findings from a shotgun metagenomic analysis of the oropharyngeal microbiome in 16 individuals with schizophrenia and 16 controls. High-level differences were evident at both the phylum and genus levels, with Proteobacteria, Firmicutes, Bacteroidetes, and Actinobacteria dominating both schizophrenia patients and controls, and Ascomycota being more abundant in schizophrenia patients than controls. Controls were richer in species but less even in their distributions, i.e., dominated by fewer species, as opposed to schizophrenia patients. Lactic acid bacteria were relatively more abundant in schizophrenia, including species of *Lactobacilli* and *Bifidobacterium*, which have been shown to modulate chronic inflammation. We also found *Eubacterium halii*, a lactate-utilizing species. Functionally, the microbiome of schizophrenia patients was characterized by an increased number of metabolic pathways related to metabolite transport systems including siderophores, glutamate, and vitamin B12. In contrast, carbohydrate and lipid pathways and energy metabolism were abundant in controls. These findings suggest that the oropharyngeal microbiome in individuals with schizophrenia is significantly different compared to controls, and that particular microbial species and metabolic pathways differentiate both groups. Confirmation of these findings in larger and more diverse samples, e.g., gut microbiome, will contribute to elucidating potential links between schizophrenia and the human microbiota.

## INTRODUCTION

Schizophrenia is a serious neuropsychiatric disorder with substantial social and economic consequences such as low performance on social knowledge and emotion recognition, high family burden, and high annual treatment costs worldwide, especially in treatment-resistant subjects (*Achim et al., 2013*; *Kennedy et al., 2014*; *Knapp, Mangalore & Simon, 2004*). Population-based studies suggest that the risk of an individual developing schizophrenia is based on both genetic and environmental factors (*van Os, Kenis & Rutten, 2010*); e.g., concordance-rates between monozygotic twins are higher than those between dizygotic twins, and adopted children of schizophrenic parents have similar risks of schizophrenia to those of their biological parents (*Fowler et al., 2012*; *Tienari et al., 1985*). Additionally, a recent genome-wide association study examining a large cohort of subjects (36,989 cases and 113,075 controls) found 108 independent associated loci that appeared to account for 7% of a given person's risk of developing schizophrenia, as assessed by polygenic risk scores. Moreover, some of the key loci were related to host immunity, thus providing genetic support for immune dysregulation in schizophrenia (*Schizophrenia Working Group of the Psychiatric Genomics Consortium, 2014*). Likewise, a population-based study has identified that single-nucleotide polymorphisms associated with schizophrenia are also associated with immunity (*Andreassen et al., 2014*), which has also been confirmed in clinical/epidemiological studies (*Benros et al., 2014*; *Eaton et al., 2010*). More studies linking immunity/immune disorders and schizophrenia have been published (reviewed in *Severance et al., 2013*); however, the source of the immune activation in most individuals has not been identified.

Recent advances in High-Throughput Sequencing (HTS) technologies have enabled the study of unculturable/difficult to culture microbial communities and have shown that the human body harbors microbial ecosystems that interact with human physiological processes, thus influencing health and disease (*Cho & Blaser, 2012*; *De Vos & De Vos, 2012*; *Pflughoeft & Versalovic, 2012*). The collection of genes and genomes belonging to these microbial communities, the microbiome, has been studied using HTS to demonstrate the extent to which the human microbiota plays a role in human health issues ranging from obesity to respiratory disease (*Huang, 2013*; *Turnbaugh et al., 2006*).

In relation to human health, observational and experimental studies have shown that the microbiome can modulate the immune response and its alterations could be associated with disease (reviewed in *Round & Mazmanian, 2009*). For instance, a recent experimental study using an osteomyelitis mice model has shown that diet can modulate the microbiome (high fat diet induces a decrease in Prevotella), which in turn offered protection against inflammatory bone disease (*Lukens et al., 2014*). Increasing

evidence suggests that alteration of the microbiota not only has effects on intestinal conditions such as Crohn's and other Inflammatory Bowel Diseases (*Gevers et al., 2014*; *Morgan et al., 2012*), but also in the development of systemic immune diseases such as rheumatoid arthritis (*Scher & Abramson, 2011*; *Wu et al., 2010*), type I diabetes (*Kriegel et al., 2011*; *Suez et al., 2014*), and allergic diseases (*Castro-Nallar et al., in press*; *Hong et al., 2010*; *Nakayama et al., 2011*; *Pérez-Losada et al., 2015*).

Although the mechanisms by which microorganisms outside the brain might impact the central nervous system (CNS) are not known, the human microbiota have been demonstrated to affect brain development and to modulate cognition through imbalances in the microbiota-gut-CNS axis (reviewed in *Clarke et al., 2012*; *Davari, Talaei & Alaei, 2013*; *Foster & McVey Neufeld, 2013*; *Hsiao et al., 2013*). As a consequence, microbiota alterations impact anxiety-like and depression-like behaviors and have been linked to neurodevelopmental disorders such as autism spectrum disorder (*Kang et al., 2013*) and multiple sclerosis (*Farrokhi et al., 2013*; *Yokote et al., 2008*).

While there is growing evidence suggesting a key role for the human microbiome in mental health, the microbiome of schizophrenia patients has not yet been extensively explored (reviewed in *Dinan, Borre & Cryan, 2014*; *Severance, Yolken & Eaton, 2014*; *Yolken & Dickerson, 2014*). Recent work by our group has shown that the oropharyngeal "phageome" of individuals with schizophrenia differs from that of non-schizophrenic controls. In particular, *Lactobacillus* phage *phiadh* was significantly more abundant in schizophrenia individuals when controlling for age, gender, race, socioeconomic status, or cigarette smoking (*Yolken et al., 2015*). However, studies evaluating the structure and diversity of bacterial and fungal communities in schizophrenia individuals are not available. In particular, gut and oral microbiome analysis could provide insights into the identity and quantity of the microbes residing in these body sites, and whether they encode functions that are relevant to and/or known to be involved in schizophrenia. In addition, the analysis of the oral microbiome could prove instrumental in the development of taxonomic and functional biomarkers because of the ease of sampling, which can be performed in a non-traumatic manner, allowing to distinguish schizophrenia patients from healthy individuals, as it has been done in other illnesses (e.g., *Farrokhi et al., 2013*).

The premise of our study is that the oropharyngeal microbiome may be associated with or contribute to an altered immune state consistent with findings in schizophrenia; hence, differences in the microbiome could be instrumental to pinpoint associations between microbial diversity and immune response. Consequently, here we aim to characterize the schizophrenia microbiome by interrogating the oropharyngeal microbiome structure regarding its taxonomic and functional diversity. This represents a second crucial step towards understanding the relationship between microbiome diversity and schizophrenia. Our initial study identified differences in the oropharyngeal phageome; whereas here we focus on the complete microbiome (virus, bacteria, fungi). Future work will investigate additional microbiome compartmentalizations, including gut, and amniotic fluid.

## MATERIALS AND METHODS

### Samples and sequencing

Participants were individuals with schizophrenia and non-psychiatric controls from the Stanley Research Program at Sheppard Pratt Hospital who were enrolled in the period between January 1, 2008 and March 1, 2012 in a study of the association between antibodies to infectious agents and serious mental illness. The methods for identification and recruitment of individuals with schizophrenia and controls have been previously described (*Dickerson et al., 2013*). Briefly, individuals with schizophrenia were recruited from psychiatric treatment programs at a large psychiatric health system and community psychiatric programs in central Maryland. Inclusion criteria were: a diagnosis of Schizophrenia or Schizoaffective disorders (established by consensus of the research team based on the SCID for DSM-IV Axis 1 Disorders—Patient Edition and available medical records). Participants met the following additional criteria: age 18–65; proficient in English; absence of any history of intravenous substance abuse; absence of mental retardation; absence of HIV infection; absence of serious medical disorder that would affect cognitive functioning; absence of a primary diagnosis of alcohol or substance use disorder (*First et al., 2012*). In order to define an individual with schizophrenia, we used the criteria from the DSM IV manual of the *American Psychiatric Association (2000)*. The Institutional Review Boards of Sheppard Pratt Hospital and the Johns Hopkins School of Medicine approved this study. Informed consent was obtained from all participants prior to enrollment into the study (Protocols SMRI/SPHS: 2002-01 and SF/SPHS: 2000-02).

We selected the oropharynx microbiome since it is easily accessible by non-invasive techniques and biological samples from cases and controls could be collected and processed in an identical manner. Moreover, studies have found the oral microbiome to be associated with immune and neurological diseases such as inflammatory bowel disease and Alzheimer's disease, and offers opportunities for development of taxonomic and functional biomarkers (*Docktor et al., 2012*; *Farrokhi et al., 2013*; *Shoemark & Allen, 2014*).

Throat swabs were obtained at the study visit (16 individuals with schizophrenia and 16 controls) by research staff that rubbed a sterile cotton swab at the back of subjects' throat and then immediately put the swab into a sterile container. The swabs were either sent immediately to the processing laboratory or refrigerated and then sent to the processing laboratory in a refrigerated container. Throat swabs were kept frozen at −70 °C until further processing. DNA was extracted from throat swabs using Qiagen's Gentra Puregene Buccal Cell Kit. The collection brush heads from the swab ends were excised and incubated at 65 °C overnight in the kit cell lysis solution. Aliquots of 75–100 ng of DNA were used to generate sequencing libraries using the Nugen Ultralow DR Multiplex System following the manufacturer's instructions. Briefly, following sonication, the DNA fragments were end repaired and ligated to barcoded adaptors. Using the adaptors as PCR priming sites, the library was amplified using 15 cycles of amplification. The samples were purified by chromatography and analyzed by capillary electrophoresis in a Bioanalyzer 2100 to confirm size and concentration. Sequencing libraries were matched one case and one control per lane and anonymized until analysis. Samples were sequenced at the same time

to further minimize batch effects. Sequences were generated using the Illumina HiSeq 2000 platform producing approximately 58-million single-end reads of 100 nucleotides in length per sample (File S1). All sequence data were deposited in the Sequence Read Archive and are available under the BioProject PRJNA255439.

## Sequence read preprocessing

Reads were preprocessed using PRINSEQ-lite 0.20.4 (trimming reads and bases <25 PHRED, removing exact duplicates, and reads with undetermined bases). We constructed a 'target' genome library containing all bacterial, fungal, and viral sequences from the Human Microbiome Project Reference Database (http://www.hmpdacc.org/reference_genomes/reference_genomes.php; 131 archeal, 326 lower eukaryotes, 3,683 viral, 1,751 bacterial species) using the PathoLib module from PathoScope 2.0 (*Hong et al., 2014*). We aligned the reads to these libraries using the Bowtie2 algorithm (*Langmead & Salzberg, 2012*), and then filtered any reads that also aligned to the human genome (hg19) as implemented in PathoMap (—very-sensitive-local -k 100—score-min L,20,1.0). We then applied PathoScope 2.0—specifically the PathoID module (*Francis et al., 2013*)—to obtain accurate read counts for downstream analysis.

## Statistical analyses

Exploratory and differential species abundance analyses were performed in R 3.1.2 and Bioconductor 3.0 (*Gentleman et al., 2004*) using packages xlsx 0.5.7, gtools 3.4.1, CHNOSZ 1.0.3.1, plyr 1.8.1, ggplot2 1.0.0, reshape2 1.4.1, gplots 2.16.0, Phyloseq 1.10.0, and DESeq2 1.6.3 (*Love, Huber & Anders, 2014*; *McMurdie & Holmes, 2013*; *Wickham, 2009*). Briefly, various indices (Richness: Observed, Chao1, ACE; richness and evenness: Shannon, Simpson, Inverse Simpson, Fisher's alpha) were obtained using the plot_richness function of the PhyloSeq package and beta diversity was obtained using R base package. We used richness indices to estimate the number of species in the microbiome with (Chao1; ACE) (*Chao, 1984*; *Colwell & Coddington, 1994*) and without (Observed) correction for subsampling. In addition, we used metrics that aim to measure diversity by accounting for "evenness" or homogeneity (Shannon; Fisher; Simpson; InvSimpson) (*Jost, 2007*). For instance, communities dominated by one species will exhibit low evenness, as opposed to communities where species are relatively well represented (high evenness).

For statistical comparison between cases and controls, the number of mapped reads estimated in PathoScope was normalized across all samples using the variance stabilized transformation method as implemented in DESeq2 using a generalized linear model (*Love, Huber & Anders, 2014*). Then, statistical inference was performed using the negative binomial Wald test (with Cook's distance to control for outliers (*Cook, 1977*)) and adjusted by applying the Benjamini–Hochberg method to correct for multiple hypotheses testing (*Benjamini & Hochberg, 1995*) at an alpha value =0.01. We controlled for covariates by adding them as coefficients in DESeq's linear model.

We also used generalized linear models implemented in STAMP and MaAsLin (*Morgan et al., 2012*; *Parks et al., 2014*) to test for differences between groups. In STAMP, differences in relative abundance between two groups of samples were compared using White's

non-parametric $t$ test (*White, Nagarajan & Pop, 2009*). We estimated confidence intervals using a percentile bootstrapping method (10,000 replications), and false discovery rate (FDR) in multiple testing was controlled by the Storey's FDR at 0.05 (*Storey, Taylor & Siegmund, 2004*). MaAsLin is a multivariate statistical framework that finds associations between clinical metadata and microbial community abundance. These associations are without the influence of the other metadata in the study. In our study, we used MaAsLin to detect the effect of schizophrenia (presence/absence) in microbiome species composition taking into account the effects of other variables (confounders) in the study population (medication, smoker, age, gender and race).

Descriptive statistics were run on all samples. Cases and controls were compared with respect to demographic and substance use variables; $\chi^y$ tests were used for categorical variables and $t$ tests were used for continuous variables. Principal coordinate analysis (PCoA) was performed on a Jensen–Shannon distance matrix derived from read counts aggregated by genus as estimated in PathoScope.

In order to explore and formally test for differences in the coding potential of the oropharyngeal microbiome, non-human reads were mapped against the Kyoto Encyclopedia of Genes and Genomes (*Kanehisa & Goto, 2000*) (KEGG; from June 2011; 1291309 genes with KO assignments) database using a two-stage local alignment algorithm as implemented in UBLAST (e-value $1e^{-9}$), part of the USEARCH package v7.0.1090 (*Edgar, 2010*). Then, metabolic pathway abundance and coverage was estimated using the Human Microbiome Project metabolic reconstruction pipeline, HUMAnN v0.99, where pathways are inferred as gene sets using maximum parsimony as the optimality criterion [MinPath (*Ye & Doak, 2009*)] and smoothed-averaged over all genes within a pathway. Significant differences between groups were tested using Kruskal–Wallis rank sum and Wilcoxon tests (alpha = 0.05) using Linear Discriminant Analysis as implemented in LEfSe (*Segata et al., 2011*). All figures were plotted using the ggplot2 and PhyloSeq packages.

# RESULTS

## Study sample demographic variables

The study sample consisted of 16 schizophrenia patients and 16 controls. Participants had a mean age of 34.5 years, were 56.3% male, and 37.5% white. On average, their mothers had over 13 years of education, and 31.3% of participants smoked. Cases were more likely to be cigarette smokers than controls ($\chi^2 = 18.6$; *p value* $< 0.0001$; 62.5% and 0%, respectively) and were also more likely to have a higher body mass index (BMI; controls = 25.5, cases = 34.7; *p value* $< 0.0001$). Groups did not differ significantly on demographic variables such as maternal education, self-reported race, age, or gender (Table 1).

## Microbial communities in the oropharynx of schizophrenia patients differ significantly from those in controls

At the phylum level, schizophrenia samples exhibit higher proportions of Firmicutes across samples in comparison to controls, where we observe higher relative proportions of Bacteroidetes and Actinobacteria (Fig. 1). Relative proportions of other phyla such as

**Table 1 Study samples' demographics data.** Cases and controls were matched and not statistically different with the exception of smoking condition and body mass index.

| | Entire sample (N = 32) | Controls (N = 16) | Schizophrenia cases (non-smoking; N = 6) | Schizophrenia cases (all; N = 16) |
|---|---|---|---|---|
| Age | 34.5 ± 7.8 | 34.3 ± 10.1 | 35.9 ± 3.4 | 34.7 ± 4.8 |
| Male gender | 18/32 (56.3%) | 9/16 (56.3%) | 3/6 (50%) | 9/16 (56.3%) |
| White race | 12/20 (37.5%) | 5/16 (31.3%) | 4/6 (66.6%) | 7/16 (43.8%) |
| Mother's education | 13.6 ± 2.9 | 14.1 ± 3.0 | 14 ± 3.4 | 13.1 ± 2.97 |
| Cigarette smoker | 10/32 (31.3%) | 0/16 (0%) | 0/0 (0%) | 10/16 (62.5%) |
| Body Mass Index | 30.1 ± 7.0 | 25.5 ± 4.5 | 33.4 ± 7.5 | 34.7 ± 6.0 |

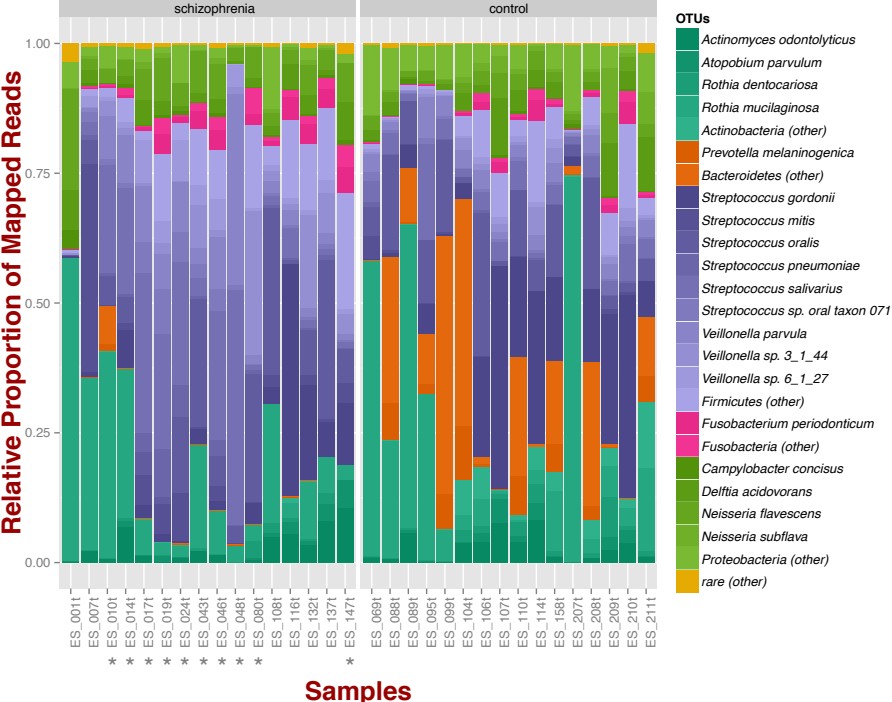

**Figure 1 Oropharyngeal microbial composition at phylum and species levels exhibits different patterns for schizophrenia and control samples.** The stacked bar chart shows the most prevalent species present in schizophrenia and controls color-coded by phylum. Green, Actinobacteria; Orange, Bacteroidetes; Blue, Firmicutes; Green, Proteobacteria. The symbol (*) indicates samples from smoker individuals.

Fusobacteria and Proteobacteria do not differ greatly (Fig. 1). Overall, groups do not differ significantly at the phylum level, which is also supported by non-metric multidimensional scaling (Fig. S1B; NMDS; Bray–Curtis dissimilarity). Differences between smoker and non-smoker cases are not evident at the phylum level (Fig. 1; smoker cases denoted with a star).

Regarding species diversity among samples, we observe that controls are richer in the number of species compared to schizophrenia samples. The median number of observed

species is higher than the interquartile range of schizophrenia samples, which is congruent with other richness metrics (Chao1, ACE, and Fisher; Fig. S1A), suggesting that controls contain a higher number of lower abundance species as opposed to schizophrenia samples. However, we observe that species abundance in controls is dominated by fewer species (*Streptococcus* spp.; Fig. 1), as evidenced by their lower evenness or homogeneity (Simpson, and Inverse Simpson; Fig. S1A), although both groups are fairly equivalent in species richness when accounting for species evenness (Shannon; Fig. S1A). Chao1 index provides an estimate of the expected number of species in a habitat. We found that the observed richness is similar to Chao1 richness, suggesting that we are capturing most of the diversity present in the samples (Fig. S1A).

## Microbial species commonly inhabiting the oropharynx are differentially more abundant in schizophrenia patients than in controls

Out of a total of 25 differentially abundant species (bacteria and fungi), we found 6 microbial species to be more abundant in cases than controls after accounting for different library sizes, smoking condition and medication (covariates were added as extra coefficients; Wald test; *p value* < 0.01). Overall, schizophrenia samples were relatively more abundant for Lactic Acid Bacteria (LAB) including *Lactobacillus* and *Bifidobacterium*. Of these, the largest effect was observed in *Lactobacillus gasseri*, which appeared to be at least 400 times more abundant in schizophrenia patients than controls ($\log_2 = 8.4 \pm 1.2$ standard error). We also detected *Eubacterium halii*, a lactate-utilizing bacterium present in human feces, and *Candida dubliniensis*, which is an opportunistic fungus that is part of the oral fungal microbiome (Table 2).

Using STAMP, we also found *Streptococcus gordonii*, *Streptococcus thermophilus*, and *Streptococcus sp.* (oral taxon 071) to be relatively more abundant in schizophrenia. The latter species was also detected by MaAsLin, which in addition detected *Bifidobacterium pseudocatenulatum*, *Bifidobacterium breve* (File S3).

We also identified species that were related to variables other than schizophrenia/non-psychiatric controls. Among these, we found that some species were related to individuals' age (*Neisseria subflava*, *Neisseria flavescens*, *Neisseria polysaccharea*, *Escherichia fergusonii*, and *Pseudomonas protegens*), being white (*Klebsiella variicola*, Actinomyces phage_Av-1, *Streptococcus sp.* (oral taxon 071)), and to cigarette smoking (*Streptococcus mitis*, *Streptococcus pneumoniae*) (File S3). Regarding species relatively more abundant in cases, we found that these preferentially belonged to Pasteurella, Neisseriaceae, and Flavobacteriaceae. We also collated the list of species found in schizophrenia patients against a list of potential contaminants published by *Salter et al. (2014)* and found no obvious contaminants (Table 2).

We also tested whether microbial composition could differentiate schizophrenia patients from controls by inferring synthetic variables that could explain the variability of the samples at the genus level (PCoA on Jensen–Shannon distance; Fig. 2). We observe that schizophrenia samples tend to group together (first three coordinates = 40%, 25%

**Table 2 Microbial species relatively more abundant in schizophrenia samples than in controls.** Effect size represents the size of the difference of schizophrenia samples over controls. The effect size as an associated standard error, and multiple comparisons were adjusted using the Benjamini–Hochberg procedure (BH).

| Effect size (log2 fold change) | Effect size standard error | p value (BH adjusted) | Phylum | Genus | Species | Description |
|---|---|---|---|---|---|---|
| 8.37 | 1.17 | 2.55E−10 | Firmicutes | *Lactobacillus* | *Lactobacillus gasseri* | Lactic acid bacterium. Member of diverse communities including gut, vaginal, and oral microbiome. Appears to be the main species of *Lactobacilli* that inhabits the human gastrointestinal tract |
| 6.81 | 0.99 | 9.61E−10 | Firmicutes | *Catenibacterium* | *Catenibacterium mitsuokai* | Phylogenetic relative of *Lactobacilli*. Found in gastrointestinal tract |
| 4.82 | 0.99 | 3.94E−05 | Firmicutes | *Eubacterium* | *Eubacterium hallii*[*] | Butyrate forming, Lactate-utilizing bacterium. Present in human feces |
| 5.71 | 1.29 | 3.13E−04 | Ascomycota | *Candida* | *Candida dubliniensis* | Opportunistic fungus. Part of the oral fungal microbiome. Present in periodontal disease |
| 2.98 | 0.80 | 4.17E−03 | Firmicutes | *Lactobacillus* | *Lactobacillus salivarius* | Lactic acid bacterium. Member of diverse communities including vaginal and oral microbiome |
| 3.79 | 1.06 | 6.30E−03 | Actinobacteria | *Bifidobacterium* | *Bifidobacterium pseudocatenulatum* | Lactic acid bacterium. Gastrointestinal tract, vagina and mouth of mammals, including humans |

**Notes.**
[*] indicates that *Eubacterium hallii* has been associated with smoking in the nasopharynx microbiome (PMID: 21188149).

and 23% of variance, respectively); however, this differentiation seems to be influenced by mental health status and in part by cigarette smoking (Figs. 2A and 2B).

## Microbial metabolic pathways differ between schizophrenics and controls

We identified 18 metabolic pathways enriched and 14 decreased in schizophrenia (*p value* < 0.05; Fig. 3). Pathways significantly associated with schizophrenia were related to environmental information processing, in particular to transport systems such as saccharide, polyol, and lipid transport systems (M00197, M00194, M00200), peptide and nickel transport (M00239), metallic cation, iron-siderophore, and vitamin B12 transport (M00246), and phosphate and amino acid transport (M00222), including glutamate transport (M00233). In turn, pathways found in control individuals, but not in individuals with schizophrenia, were related to energy metabolism, such as ATP synthesis and ATP synthase (M00144, M00150, M00157, M00164) and carbohydrate and lipid metabolism, such as central carbohydrate metabolism (M00011, M00009, M00007) and lipopolysaccharide metabolism (M00060).

## DISCUSSION

This study is among the first surveys on the composition and differences in the microbiome of schizophrenia patients and healthy controls using shotgun metagenomic sequencing.
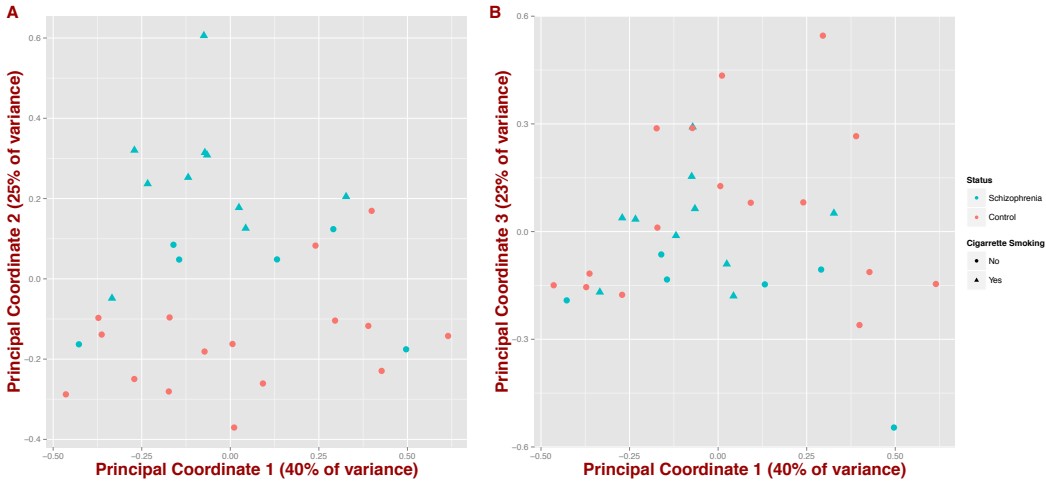

**Figure 2 Covariation of community structure shows that diversity patterns of samples correlate with disease status, i.e., schizophrenia and controls, and potentially with smoking (at the genus level).** Points represent principal coordinate analysis (PCoA loadings) on Jensen–Shannon Diversity distances. Principal coordinates 1 and 2 in (A) (65% of variance) and principal coordinates 1 and 3 in (B) (63% of variance).

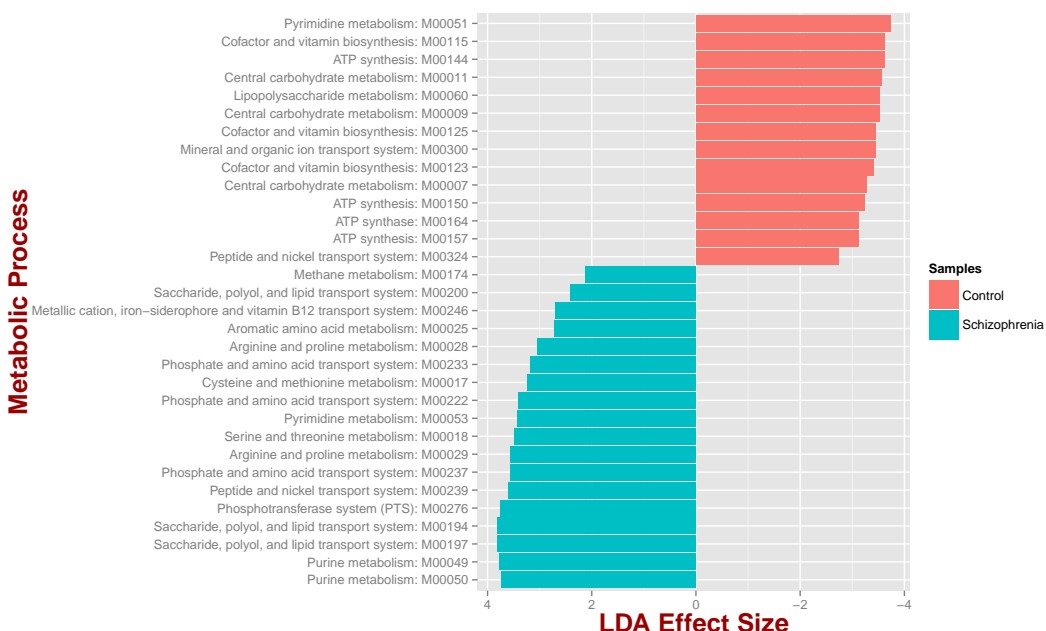

**Figure 3 Microbial metabolic pathways with significantly altered abundances in the schizophrenia oropharyngeal microbiome.** MXXXXX codes correspond to KEGG modules, i.e., a collection of manually defined functional units (genes). LDA, linear discriminant analysis.

The oropharyngeal microbiome is particularly attractive for microbiome-associated biomarker development because biological samples can be collected and processed in an identical and non-traumatic manner from both individuals with psychiatric disorders and controls. Additionally, while oral and gut microbiomes share little taxonomic resemblance,

both are significantly associated (*Ding & Schloss, 2014*), which might be instrumental for future diagnostic aids. In addition, shifts in oral microbiome diversity have been linked to immune and neurological diseases such as inflammatory bowel disease and Alzheimer's disease (*Docktor et al., 2012*; *Shoemark & Allen, 2014*). Although there is little mechanistic understanding about the role of the microbiome on inflammatory and neurological disease, some studies have found specific markers (microbes and metabolites) associated with both gut and oral microbiomes. These markers have been shown to enter the systemic circulation and elicit systemic immune responses, thus serving as specific biomarkers of disease (*Clark et al., 2013*; *Farrokhi et al., 2013*).

We show that the oropharynx microbiome in schizophrenics is significantly different in comparison from that of healthy controls. High-level differences were evident at both the phylum and genus levels; however, the overall composition of the microbiome is similar to those previously reported (*Belda-Ferre et al., 2011*; *Dewhirst et al., 2010*). Proteobacteria, Firmicutes, Bacteroidetes, and Actinobacteria dominated both schizophrenia patients and controls, with Ascomycota being more abundant in schizophrenia patients than controls. At the genus level, we identified key members of the healthy oropharynx microbiome such as *Prevotella*, *Capnocytophaga*, *Campylobacter*, *Veillonella*, *Streptococcus*, *Neisseria* and *Haemophilus*; as reported in other studies that characterize the oropharynx microbiome (*Charlson et al., 2010*). We also detected in schizophrenia samples an increase in lactic acid bacteria, *Candida*, and *Eubacterium*, and a marked reduction of *Neisseria*, *Haemophilus*, and *Capnocytophaga*, suggesting a characteristic dysbiosis signature (*Gao et al., 2014*). Notably, decreases in *Neisseria* and *Capnocytophaga* abundance have been associated with cigarette smoking (*Charlson et al., 2010*); therefore, although we incorporate cigarette smoking as a coefficient in our models, we can not rule out that this observation is due to smoking and not mental health status. We also found that *Streptococcus mitis* and *Streptococcus pneumoniae* abundance was associated with smoking, which is consistent with other findings from culture independent (*Charlson et al., 2010*) and culture-based studies (*Brook & Gober, 2005*).

Reagent and laboratory contamination in genomics in general (*Merchant, Wood & Salzberg, 2014*; *Strong et al., 2014*) and in microbiome studies in particular (*Lusk, 2014*; *Salter et al., 2014*; *Weiss et al., 2014*) have been recently brought into attention as an alarming problem. Although contamination affects primarily samples coming from low diversity ecosystems, in our analysis we used high statistical significance (*p value* < 0.01 coupled with independent filtering and outlier removal) to prevent spurious detection of contaminants as differentially abundant. Moreover, our laboratory protocols (library preparations; sequencing) were performed simultaneously for all samples, and both case and control samples were loaded in the same sequencing lane, thus minimizing chances of contamination, i.e., contamination correlated with disease status.

Some of the most abundant species found in samples from individuals with schizophrenia (e.g., *Lactobacillus*, *Bifidobacterium*, *Candida*) have been reported previously as members of the respiratory tract microbiome and have been linked to opportunistic infections (*Kim & Sudbery, 2011*; *Land et al., 2005*). However, relatively abundant species

in schizophrenia samples are not exclusive inhabitants of the respiratory tract, as bacterial species we detected have also been described in the gastrointestinal tract (*Dillon, 1998*; *Duncan, Louis & Flint, 2004*; *Kageyama & Benno, 2000*). In addition, some reports show that oral and intestinal *Lactobacilli* share a common origin in the oral cavity (*Dal Bello & Hertel, 2006*), suggesting that species such as *L. salivarius* and *L. gasseri* (both identified in this study) might be allochthonous to the human gut and autochthonous to the respiratory tract (*Reuter, 2001*), and therefore potentially useful for microbiome signature development (*Aagaard et al., 2012*; *Haberman et al., 2014*). In addition, *Lactobacilli* (except *L. salivarius*) in the intestine has been linked to emotional behavior regulation and alteration of mRNA expression patterns of the GABA receptor via the vagus nerve (*Bravo et al., 2011*). *Lactobacilli* have also been linked to chronic inflammation and anxiety-like behavior (*Bercik et al., 2010*). In particular, *L. gasseri* has been shown to modulate the immune system by altering the function of dendritic cells, enterocytes, and components of innate immunity (*Luongo et al., 2013*; *Selle & Klaenhammer, 2013*).

We also detected non-bacterial species such as fungi and DNA viruses (phage; File S2) that would have been beyond the scope of gene marker surveys, e.g., 16S rRNA gene (*Weinstock, 2012*). One fungal species, *Candida dubliniensis*, was differentially abundant in individuals with schizophrenia, which has been associated to immune-compromised individuals (*Eggimann & Pittet, 2014*; *Sebti et al., 2001*), but also found in healthy respiratory tract microbiomes (prevalence of 75% *Candida*) (*Ghannoum et al., 2010*). This suggests that the high abundance of these microbes in schizophrenia individuals might be associated with altered immune responses or changes in the local environment that enable their outgrowth, as observed in other diseases (*Bisgaard et al., 2007*; *Molyneaux et al., 2013*). We also detected viruses, primarily phages, in our datasets; however our group has published a more comprehensive characterization of them in a larger study population (see *Yolken et al., 2015*).

We also assessed the functional diversity of the oropharynx microbiome in schizophrenia and control samples. There is little information regarding the functional diversity of the respiratory tract in general, and a complete lack of information in schizophrenia. In our study, we identified metabolic pathways significantly associated with the microbiomes of schizophrenics (i.e., environmental information processing) including glutamate transport (M00233), while the microbiomes of healthy controls were enriched for bioenergy pathways. Microbial metabolic pathways found in schizophrenia samples have also been described in samples from the respiratory and digestive tracts such as in gingiva, oral cavity, and stool (M00239, M00276, M00200, M00028, M0029), reinforcing the notion that oral and gut microbial communities share features and are associated (*Meehan & Beiko, 2012*; *Sczesnak et al., 2011*; *Seekatz et al., 2014*; *Segata et al., 2012*). Dominant hypotheses for the pathophysiology of schizophrenia, as well as recent genetic studies, point to neurotransmitter disturbances in glutamatergic and dopaminergic activities (*Falkai, Schmitt & Cannon, 2011*; *Schizophrenia Working Group of the Psychiatric Genomics Consortium, 2014*). While we found glutamate related pathways to be more abundant in schizophrenia samples, the design and scope of this study does not allow us to evaluate

mechanistic hypotheses regarding pathophysiology of schizophrenia and the potential contribution of the microbiome. Study designs based on metatransciptomics or dual RNAseq strategies coupled with advanced statistical techniques such as feature reduction might be instrumental to detect associations between known host genetic determinants and their expression, and microbiota structure (*Morgan et al., 2015*). Undoubtedly, study designs using larger sample sizes and both RNA and DNA information (*Franzosa et al., 2014*) collected from different body sites (e.g., intestines) are needed for accurate statistical validation of microbial and metabolic as promising biomarkers for schizophrenia. Further analyses of other body sites involving larger study populations will be presented elsewhere as we collect relevant samples.

In summary, by using metagenomic sequencing, we have shown that it is possible to distinguish schizophrenia patients from controls by profiling the oropharyngeal microbiome based on the diversity and composition of microbes. Additionally, microbiomes from schizophrenics and controls differ in the functions they potentially encode suggesting it may be important to further characterize other body sites such as the intestines. These differences could be exploited for the development of biomarkers and ultimately for therapeutic interventions.

The fact that all controls were non-smokers, although statistically accounted for in our inferences, might confound the effects of schizophrenia from those of smoking on microbiome composition. A recent study found no significant differences in diversity (alpha and beta) between the oral microbiome in healthy smokers and non-smokers without psychiatric disorders (*Morris et al., 2013*). However, this and other studies have found specific taxa (*Haemophilus influenzae*, *Streptococcus pneumoniae*, *Megasphaera* and *Veillonella* spp) to be differentially abundant in smokers and non-smokers (in humans *Charlson et al., 2010*; in mice *Voss et al., 2015*). Here we found that the distributions of two *Streptococcus* species were better explained by whether the individual was a smoker than whether the individual had schizophrenia.

Additionally, studies in animal models have indicated that changes in the microbiome can cause alterations in behavior and cognitive functioning and that these changes can be modulated by probiotic and antibiotic interventions (*Jakobsson et al., 2010*; *Martin et al., 2008*). The establishment of a link between the microbiome and behavioral and cognitive functioning in humans might lead to the development of new strategies for the prevention, management, and treatment of psychiatric disorders.

## ACKNOWLEDGEMENTS

We thank The Stanley Medical Research Institute for their support of this work. We also thank the GWU Colonial One High Performance Computing Cluster for computational time.

### Funding

This work was supported by a NIMH P50 Silvio O. Conte Center at Johns Hopkins (grant# MH-94268). ECN was funded by "CONICYT + PAI/ CONCURSO NACIONAL APOYO AL RETORNO DE INVESTIGADORES/AS DESDE EL EXTRANJERO, CONVOCATORIA 2014 + FOLIO 82140008." MP-L was funded in part by a K12 Career Development Program 5 K12 HL119994 award. KAC was supported in part by Award Number UL1TR000075 from the NIH National Center for Advancing Translational Sciences. The study contents are solely the responsibility of the authors and do not necessarily represent the official views of the National Center for Advancing Translational Sciences or the National Institutes of Health. The funders had no role in study design, data collection and analysis, decision to publish, or preparation of the manuscript.

### Grant Disclosures

The following grant information was disclosed by the authors:
NIMH P50 Silvio O.Conte Center at Johns Hopkins: # MH-94268.
CONICYT + PAI/ CONCURSO NACIONAL APOYO AL RETORNO DE INVESTIGADORES/AS DESDE EL EXTRANJERO, CONVOCATORIA 2014 + FOLIO: 82140008.
Career Development Program: 5 K12 HL119994.
NIH National Center for Advancing Translational Sciences: UL1TR000075.

### Competing Interests

Keith A. Crandall is an Academic Editor for PeerJ. Eduardo Castro-Nallar and Keith A. Crandall have a combination of ownership of, and employment in, Aperiomics, Inc. Jennifer R. Schroeder is an employee of Schroeder Statistical Consulting LLC.

### Author Contributions

- Eduardo Castro-Nallar conceived and designed the experiments, analyzed the data, contributed reagents/materials/analysis tools, wrote the paper, prepared figures and/or tables, reviewed drafts of the paper.
- Matthew L. Bendall and Marcos Pérez-Losada analyzed the data, contributed reagents/materials/analysis tools, prepared figures and/or tables, reviewed drafts of the paper.
- Sarven Sabuncyan, Emily G. Severance and Faith B. Dickerson conceived and designed the experiments, performed the experiments, reviewed drafts of the paper.
- Jennifer R. Schroeder analyzed the data, contributed reagents/materials/analysis tools, reviewed drafts of the paper.
- Robert H. Yolken conceived and designed the experiments, performed the experiments, contributed reagents/materials/analysis tools, reviewed drafts of the paper.
- Keith A. Crandall conceived and designed the experiments, contributed reagents/materials/analysis tools, reviewed drafts of the paper.

## Human Ethics

The following information was supplied relating to ethical approvals (i.e., approving body and any reference numbers):

The Institutional Review Boards of Sheppard Pratt Hospital and the Johns Hopkins School of Medicine approved this study. Informed consent was obtained from all participants prior to enrollment into the study.

## DNA Deposition

The following information was supplied regarding the deposition of DNA sequences:

All sequencing data were deposited in the Sequence Read Archive in GenBank and are available under the BioProject PRJNA255439.

## Supplemental Information

Supplemental information for this article can be found online at http://dx.doi.org/10.7717/peerj.1140#supplemental-information.

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
