# Peer review of "Composition, taxonomy and functional diversity of the oropharynx microbiome in individuals with schizophrenia and controls"

_PeerJ, doi:10.7717/peerj.1140_

## Round 0.1 · original submission · Major Revisions

· Academic Editor

Major Revisions

The reviewers provided several points that should be addressed to help guide the message being presented here. The assessments led to a categorization of the manuscript in the “Major Revisions” category. I am basically siding with the reviewers in that the premise for this study did not appear clear, and mostly appeared to be a preliminary investigation which may require further follow-up.

I was somewhat confused in questioning how this manuscript may significantly differ from the other journal paper which is implicating a possible role of phage? Co-authors EG Severance, FB Dickerson, and RH Yolken are authors on this paper and published just before submission of this manuscript; see Schizophr Bull. 2015 Feb 9. Metagenomic Sequencing Indicates That the Oropharyngeal Phageome of Individuals With Schizophrenia Differs From That of Controls.

If there are other avenues of investigation being conducted, why are they not included in the presentation and discussion? I would encourage the authors to re-evaluate the message being presented here and perhaps expand on how it relates to the big picture. Please also consider the comments provided by the reviewers and try to present a more unified front by including more details.

Reviewer 1 ·

Basic reporting

No Comments - see below.

Experimental design

No Comments - see below.

Validity of the findings

No Comments - see below.

Additional comments

The manuscript by Castro-Nallar and colleagues describes an observational study comparing the oropharyngeal microbiome of 16 individuals with schizophrenia with 16 control individuals. Microbial communities of each individual were characterized using shotgun metagenomic sequencing allowing the authors to compare the taxonomic and functional diversity of the communities. They found differences in both the taxonomic and functional diversity between the two study populations. Overall the paper is well written and the methods are appropriate but a number of details are lacking that are important in order to evaluate the validity and significance of their findings. Below I have highlighted some of the most obvious omissions of the authors.

There are numerous confounding variables specific to the study population not discussed by the authors. In particular, co-morbidity in psychiatric patients is commonplace – what other afflictions did these patients have that could also play a role in microbiome dysbioses? How were those accounted for in your models? What is the BMI (or other indication of general health) of each patient? Also, hygiene in schizophrenic patients is notoriously poor. Did you account for oral hygiene/health?

Secondarily, many of the medications taken by study subjects have side effects that could play a role in the observed dysbioses. For example, some of the medications cause dry mouth while others cause hypersalivation. Furthermore, most of the medications cause metabolic disorders that lead to obesity and type-2 diabetes. How do you know the observed differences are at all related to schizophrenia and not related to certain medications? The authors state that medication use was accounted for in the models but this is not clear how?

Generally, it is unclear how you accounted for smoking and medication in your statistical models. This is a major flaw as currently presented and must be clarified prior to publication.

Specific comments:

Lines 54-56: I am uncomfortable with the statement that the oral and gut microbiome are predictive of each other. Not all studies have observed this phenomenon and when it was observed it was specific to particular oral regions. There is a tremendous amount of spatial variability within the oral microbiome.

Line 72: Were samples self-collected? What was the timing of collection? Please provide more details as activities such as oral hygiene and eating could temporarily influence the oral microbiome.

Line 128: ‘deconfounding’ – how? Please provide more details.

Lines 203-205: By definition, doesn’t principal coordinate 1 always explain the greatest amount of variance in the data? I am not sure what you mean in this paragraph?

Line 205: What is meant by ‘phenotype’ in this sentence?

Lines 280-288: Most of this paragraph is just restating the results. Please re-work to provide some interpretation of the observed patterns.

The color scheme in Figure 1 should be reworked. Also, please identify which individuals are smokers since this is an important factor driving some of the observed differences.

The caption of Figure 2 is inadequate; please provide more detail with specific reference to each sub-panel. Separation in Figure 2A seems to be driven primarily by smoking, regardless of mental health status.

Other demographic features (like BMI or co-morbidity) could be added to Table 1.

Some of the references are incomplete.

Reviewer 2 ·

Basic reporting

The introduction covers a lot of ground without much detail. It is unclear if the references cited are reviews or original work. For example, the first sentence says there are substantial social and economic consequences without saying what they are. The second sentences starts with ‘Numerous studies’ but cites two references, and the third sentence makes a sweeping statement without evaluating the quality of the evidence. This is consistent problem throughout; the reader is not told if the evidence is from animal models or human studies nor is there any evaluation of the quality of the evidence. The justification for screening the oral microbiome rather than the gut microbiome could be strengthened in the introduction or the material justifying that choice included in the discussion moved earlier.

The discussion might more clearly place these results in the context of the literature; I would have enjoyed some discussion of whether the functional pathways identified fit in any way with what is known about schizophrenia pathogenesis.

Experimental design

Methods: the authors should at least briefly describe methods for identification and recruitment of participants; most critical what was the definition used for schizophrenia and control selection. Please include a description of the controls included in sequencing and the QA/QC procedures.

Validity of the findings

None of the controls were cigarette smokers. The oral microbiome is known to vary significantly by cigarette smoking (saliva, oral washes and samples from crevicular fluid), therefore there were only 6 informative samples from cases. The descriptive analysis should be displayed using this sub-set as it is misleading to include smokers with the case group (figures 1, 2, 3 and Table 2). The paper cited by Morris in the discussion as support for their analysis plan in fact does not support their contention: Morris et al. states that the mouth microbiome (from oral washes) differs among smokers and non-smokers but that found in samples from the lungs (from bronchoscopic alveolar lavages) did not. It seems quite likely that oral washes are more similar to throat swabs than bronchoscopic alveolar lavages.

The authors do not state whether the analysis of functional pathways was limited to non-smokers; it is unclear whether the pathways identified are related to smoking or to schizophrenia.

---

## Round 0.2 · accepted · Accept

· Academic Editor

Accept

This version is much improved from the initial submission, and builds a premise upon which the research is conducted. The earlier version seemed to miss this; and the mention of the parallel works investigating other microbiota helped to put this work in context. As a survey paper it does not force any conclusions, but simply makes observations of differences noted and suggests the trends which would be looked for in larger studies to determine the true impacts that the microbiome may relate. I would presume that the microbiome state is further complicated by other lifestyle traits that would be associated with a diagnosed condition; however, as a survey it presents a starting point for building further validation studies. I believe you sufficiently addressed the reviewer concerns and helped make your case by pointing to other related works that influenced the observations and sufficiently pointed to the methodology used. I did not note any needed revisions, was quite pleased with this version, and recommend this to be further moved forward for publication. As further microbiome studies are adapted to survey conditional states using genome skimming protocols, a higher bar may evolve and be demanded by the research community to encompass the complex relationships that exist in environment interactions.